# Synthesis and Identification of Epoxy Derivatives of 5-Methylhexahydroisoindole-1,3-dione and Biological Evaluation

**DOI:** 10.3390/molecules26071923

**Published:** 2021-03-30

**Authors:** Kariny B. A. Torrent, Elson S. Alvarenga

**Affiliations:** Department of Chemistry, Universidade Federal de Viçosa, Viçosa, MG 36570-900, Brazil; karinybragatto@gmail.com

**Keywords:** cyclic imide, theoretical calculations, DFT, CP3, DP4, herbicide, phytotoxic

## Abstract

Cyclic imides belong to a well-known class of organic compounds with various biological activities, promoting a great interest in compounds with this functional group. Due to the structural complexity of some molecules and their spectra, it is necessary to use several spectrometric methods associated with auxiliary tools, such as the theoretical calculation for the structural elucidation of complex structures. In this work, the synthesis of epoxy derivatives of 5-methylhexahydroisoindole-1,3-diones was carried out in five steps. Diels–Alder reaction of isoprene and maleic anhydride followed by reaction with *m*-anisidine afforded the amide (**2**). Esterification of amide (**2**) with methanol in the presence of sulfuric acid provided the ester (**3**) that cyclized in situ to give imides **4** and **4-ent**. Epoxidation of **4** and **4-ent** with *meta*-chloroperbenzoic acid (MCPBA) afforded **5a** and **5b**. The diastereomers were separated by silica gel flash column chromatography, and their structures were determined by analyses of the spectrometric methods. Their structures were confirmed by matching the calculated ^1^H and ^13^C NMR chemical shifts of (**5a** and **5b**) with the experimental data of the diastereomers using MAE, CP3, and DP4 statistical analyses. Biological assays were carried out to evaluate the potential herbicide activity of the imides. Compounds **5a** and **5b** inhibited root growth of the weed *Bidens pilosa* by more than 70% at all the concentrations evaluated.

## 1. Introduction

In 2050, it is estimated that the global population will reach the mark of 9.1 billion people. As a result, world food production will need to increase by 70–100%. Weeds are agricultural pests and cause the highest percentage of loss of income from food production. They compete with the plantations for sunlight, water, and nutrients, harbor insects and pathogens (fungi, bacteria, and viruses), promoting losses in production. In addition, weeds destroy native habitats, threatening plants and animals in the local ecosystem [1,2].

Cyclic imides belong to an important class of organic compounds, being particularly important in synthetic, pharmacological, and industrial chemistry. Imides are compounds with high biological potential because they are electrically neutral and hydrophobic, being able to cross biological membranes [3,4,5]. Among its biological activities, herbicidal activity stands out and some commercial herbicides are presented in Figure 1 [6].

Spectroscopic and spectrometric techniques are tools used in the structural elucidation of new substances, and nuclear magnetic resonance (NMR) spectroscopy is frequently used. However, in some situations, the data obtained by these techniques are not sufficient for structural elucidation of organic compounds with complex structures, or even due to overlapping signals in the ^1^H NMR spectra, requiring the use of other tools, such as computational chemistry [7,8,9].

Structural elucidation of molecules using computational chemistry, chemical shifts, and spin–spin couplings is calculated using the functional density theory (DFT) and considering solvation effects and relativistic effects. Calculated data are compared to those obtained experimentally through statistical methods to establish the structure of the molecule [10,11,12].

Statistical methods commonly used in the structure determination of organic compounds are mean absolute error (MAE), CP3, and DP4 probabilities. DP4 tool is applied when there are experimental data for one compound, which is compared to two or more candidate structures, whereas CP3 is used when NMR chemical shifts are available for two or more substances. The calculation of DP4 can be done at http://www.jmg.ch.cam.ac.uk/tools/nmr/DP4 (accessed on 28 January 2021) for assigning one set of experimental data to one of many candidate structures. The calculation of CP3 can be done at http://www-jmg.ch.cam.ac.uk/tools/nmr/CP3.html (accessed on 28 January 2021) for assigning two or more set of experimental data to one of many potential structures. The probability that the assignment combination is correct (**5a**_exp_ = **5a**_calc_, **5b**_exp_ = **5b**_calc_) is calculated by the Bayes’ theorem (Equation S1) [13,14,15,16].

Therefore, we decided to use the known Diels–Alder reaction to prepare the tetrahydrophthalimide (**4**) in three steps, which was converted into the epoxy compounds (**5a** and **5b**). From this reaction, we obtained two novel pairs of diastereomers whose relative configurations were determined by comparison of the experimental with the computed NMR chemical shifts.

In this context, we prepared two novel epoxy compounds derived from 5-methylhexahydroisoindole-1,3-dione from the Diels–Alder reaction between maleic anhydride and isoprene. The diastereoisomers had their structures elucidated by NMR, and their relative configurations were confirmed by comparing the experimental with the calculated NMR chemical shifts. The herbicidal activities of these compounds were evaluated against seeds of *Bidens pilosa* L., *Cucumis sativus* L., *Lactuca sativa* L., and *Sorghum bicolor* L.

## 2. Results and Discussions

### 2.1. Synthesis

Diels–Alder reaction of isoprene and maleic anhydride followed by nucleophilic addition of *m*-anisidine afforded the *cis*-carbamoyl-carboxylic acid (**2**). The *cis* geometry of compound (**2**) was directly influenced by the stereoselectivity of the Diels–Alder reaction. The *cis*-carbamoyl-carboxylic acid (**2**) was identified as a mixture of constitutional isomers by spectrometric methods. Treating compound (**2**) with sulfuric acid in methanol afforded methyl *cis*-carbamoyl carboxylate (**3**), which was identified in the presence of compound (**4**). The relative quantities of compounds **3** and **4** in the mixture were determined by gas chromatography with flame ionization detection (GC-FID, Appendix A in the Supporting Information (SI)). Sample peak areas were integrated to furnish 63% of imide (**4**) and 37% of methyl *cis*-carbamoyl carboxylate (**3**). Compounds **3** and **4** were not separated, before reaction with *meta*-chloroperbenzoic acid (MCPBA) to afford (**5a** + **5a-ent**) and (**5b** + **5b-ent**). Hereafter, we refer to the enantiomeric mixtures (**5a** + **5a-ent**) and (**5b** + **5b-ent**) as (**5a**) and (**5b**), respectively. Compounds (**5a**) and (**5b**) were obtained in 85% yield after silica-gel column chromatography (Scheme 1).

#### 2.1.1. Structure Identification of Compound (**5b**)

Compound (**5b**) was isolated as the major product in 66% yield by silica-gel column chromatography. The singlets (s) at *δ* = 1.32 and 3.80 ppm observed in the ^1^H NMR were assigned to the methyl and methoxy groups, respectively. The triplet (t) at *δ* = 6.84 ppm with *J* = 2 Hz was assigned to H2′ due to coupling with H4′ and H6′. The doublet of doublets of doublets (ddd) at *δ* = 6.84 and 6.92 ppm (*J* = 8.0, 2.0, and 1.0 Hz) were assigned to H6′ and H4′, respectively. The most deshielded signal, at *δ* = 7.36 ppm (t, *J* = 8.0 Hz), was assigned to H5′ (Figure 2). The doublet at *δ* = 3.04 ppm (*J* = 4.0 Hz) was assigned to H6a due to coupling with H6x (dihedral angle of 42°). The absence of coupling of H6a with H6y can be explained by their dihedral angle of 75°. According to the Karplus equation, a dihedral angle of 75° should provide coupling constants of 0–1 Hz. The chemically equivalent and magnetically different hydrogens H2a and H5a were assigned to the non-first order multiplet at *δ* = 2.89 ppm. The signal at *δ* = 2.22 ppm (dd) was assigned to H6y, which was coupled to the geminal H6x (*J* = 15.0 Hz) and the vicinal H5a (*J* = 7.0 Hz). The signal at *δ* = 2.18 ppm (dd) was assigned to H2y, which was coupled to H2x (*J* = 15.0 Hz) and H2a (*J* = 7.0 Hz). COSY (Correlation Spectroscopy) contour map (Appendix A) was of great value to assist in these assignments. The nuclear Overhauser effect (NOE) of H2x/H6x observed in the NOESY (Nuclear Overhauser Effect Spectroscopy, Appendix A) was used to confirm the distance separating these cross-relaxing hydrogens and, by extension, the relative stereochemistry of H2y/H6y.

After the assignment of the hydrogens by interpretation of the ^1^H NMR, COSY, and NOESY spectra, the corresponding carbons were assigned using HMQC (Heteronuclear Multiple Quantum Coherence, Appendix A) and HMBC (Heteronuclear Multiple Bond Coherence, Appendix A) cross-correlations. The experimental NMR chemical shifts and a complete assignment of the signals for compound **5b** are shown in Table 1.

#### 2.1.2. Structure Identification of Compound (**5a**)

Compound (**5a**) was isolated as the minor product in 19% yield by silica-gel column chromatography. The singlets (s) at *δ* = 1.40 and 3.80 ppm observed in the ^1^H NMR were assigned to the methyl and methoxy groups, respectively. The aromatic hydrogens H2′, H4′, H5′, and H6′ were located at the deshielded region *δ* = 6.79–7.36 ppm, and their assignments were done according to their chemical shifts and couplings. The signals of H2a, H5a, and H6a were superimposed as a multiplet at *δ* = 2.98–3.20 ppm (Figure 2). The cross-correlation with C2a, C5a, and C6a, observed in the HMQC (Appendix A), confirmed the assignments of H2a, H5a, and H6a to the multiplet at *δ* = 2.98–3.20 ppm in the ^1^H NMR. The doublet of doublets at *δ* = 2.08 and 2.68 ppm were assigned to H6y and H6x, respectively. The multiplicity of H6y was due to geminal coupling with H6x (*J* = 15.0 Hz) and vicinal coupling to H5a (*J* = 10.0 Hz). The hydrogens H2y and H2x were assigned to the doublet of doublets at *δ* = 2.01 and 2.50 ppm, respectively. The relative stereochemistry of **5a** was confirmed by the cross-correlations of H2a/H2x, H5a/H6x, and H6y/H6a observed in the NOESY (Appendix A).

Table 2 presents nuclei dihedral angles and coupling constants predicted by Karplus equation [17] for compounds **5a** and **5b**. The dihedral angles were visualized in GaussView 6.0 [18] for the most stable conformer after structure optimization using Gaussian 16 [18]. NOESY and coupling constants data were used to define the relative stereochemistries of compounds **5a** and **5b**.

The carbon atoms of compound **5a** were fully assigned using DEPT (Distortionless Enhancement by Polarization transfer, Appendix A), HMQC (Appendix A), and HMBC (Appendix A) spectra. ^1^H and ^13^C NMR spectra of compounds **5a** and **5b** are presented in Figure 2.

Table 3 summarizes the experimental NMR data for compound **5a**, coupling constants, and cross-correlations observed in the COSY and NOESY spectra.

### 2.2. Computational Analysis

All ^1^H and ^13^C NMR signals were assigned for candidate structures **5a** and **5b** by interpretation of the NMR spectra. The diastereomers were further analyzed by comparing the calculated chemical shifts for each candidate structure with their experimental data. 

The geometries of the conformers, identified by an initial molecular mechanics conformational search [19], for each of **5****a** and **5b** were optimized at the M06-2X/6-31+G(d,p) level of theory and then subjected to NMR chemical shift calculation using the B3LYP (Becke, 3-parameter, Lee–Yang–Parr) functional and 6-311+G(2d,p) basis set. The shifts for each diastereomer were Boltzmann averaged according to the M06-2X energies of the full set of conformers. The experimental ^1^H and ^13^C NMR chemical shifts for **5a** and **5b** and the linearly corrected computed shifts are shown in Appendix A.

The corrected mean absolute error (CMAE) and the statistical analyses CP3 and DP4 were then used to identify the better match between the experimental and computed data sets.

Linear regressions of the ^1^H and ^13^C NMR chemical shifts of (**5a_exp_** + **5b_exp_**) **vs.** (**5a_calc_** + **5b_calc_**) and (**5a_exp_** + **5b_exp_**) vs. (**5****b_calc_** + **5a_calc_**) were carried out, as shown in Appendix A. The chemical shifts after linear correction were compared with the experimental data to calculate the CMAE.

Incorrect matching **5b_exp_**
_≠_
**5a_calc_** for the ^1^H NMR chemical shifts presented larger CMAE (0.10 ppm) than the correct matchings **5a_exp_** = **5a_calc_** and **5b_exp_** = **5b_calc_** (0.04 ppm). The correct matching for the ^13^C NMR chemical shifts **5a_exp_** = **5a_calc_** and **5b_exp_** = **5b_calc_** presented a smaller CMAE value than the incorrect matching (1.66 vs. 1.91). The correct matchings are highlighted in green in Table 4. 

The statistical approach CP3 [14,20] without linear correction and assignment of the signals provided 100% probability for **5a_exp_** = **5a_calc_** and **5b_exp_** = **5b_calc_** matching using ^13^C NMR chemical shifts. The results reported by CP3 were obtained by comparing differences in calculated data with differences in experimental chemical shifts for equivalent nuclei. CP3 calculations were performed by transferring the NMR data to the URL located at http://www-jmg.ch.cam.ac.uk/tools/nmr/CP3.html (accessed on 28 January 2021). Structure assignment was done using chemical shifts obtained in single-point calculations on geometries optimized at M06-2X/6-31+G(d,p) level of theory. The advantage of CP3 over CMAE is that it does not need linear regression and correction of the chemical shifts. Another major advantage of CP3 is that it can be carried out without assignment of the NMR signals, unlike CMAE, which needs NMR interpretation by experts for the assignment of the signals.

DP4 [13,15,20] is applied when only one set of experimental chemical shifts is available, unlike CP3, which applies to the situation of assigning a pair of diastereoisomers when more than one experimental data sets are available. Therefore, DP4 complements the probabilities obtained by CP3. 

The calculated chemical shifts for candidate structures **5a** and **5b** were compared with the experimental data for compound named **5a** using the DP4 probability method. These values were transferred to the web at http://www-jmg.ch.cam.ac.uk/tools/nmr/DP4/ (accessed on 28 January 2021), and the DP4 probability value was calculated automatically. DP4 analysis without linear correction and without assignment of the signals provided 100% probability for **5a_exp_** = **5a_calc_** and 93.9% matching for **5b_exp_** = **5b_calc_** using ^1^H and ^13^C NMR chemical shifts. The DP4 and CP3 statistical probabilities are shown in Table 4.

### 2.3. Biological Assay

The biological tests were carried out with seeds of three dicotyledonous plants (lettuce, cucumber, and beggartick) and one monocotyledonous plant (sorghum).

The results of the bioassays were presented in bar graphs with their respective standard deviation values. The values of inhibition and stimulation of the substances were evaluated according to the growth (positive values) or inhibition (negative values) of the root and stem of the tested seeds. Dual (Syngenta^®^ Company, São Paulo, Brazil; S-metolachlor, Figure 3) was used as positive control and an aqueous solution of 0.3% DMSO (*v*/*v*) as a negative control.

Assessing the development of lettuce seeds (Figure 4), all substances interfered with the growth of the aerial and root parts of the seeds; however, the results observed for the root were more satisfactory. In general, substance **5a** presented better inhibition results than substance **5b**. At the concentration of 500 μM, substance **5a** presented inhibition of 43% of the aerial and 52% of the root parts. At the concentration of 100 μM, substance **5a** was the most active, even compared with the positive control, with 69% roots inhibition.

Assessing the development of the aerial and root parts of cucumber seeds (Figure 5), all substances interfered with their development; however, none of the tested substances presented satisfactory results compared with the commercial herbicide.

Evaluating the development of sorghum seeds (Figure 6), all substances interfered with the growth of the aerial and root parts of the seeds; however, substance **5a** presented better inhibition results, in general, compared with substance **5b**. At the concentration of 500 μM, substance **5a** presented close to 50% inhibition of stem and root parts of sorghum. At 150 μM, substance **5a** presented more than 55% inhibition of the roots and stems—results superior to that presented by the commercial herbicide at the same concentration.

Assessing the development of the weed *Bidens pilosa* (beggartick) seeds, both epoxides affected their development (Figure 7). Epoxide **5a** presented better inhibition of the stalk and root parts of beggartick than **5b** at the concentration of 500 μM. At 500 μM, **5a** and **5b** presented 86% and 81% inhibition of the root parts of beggartick, respectively. The herbicide activity, displayed by **5a** and **5b,** was comparable with those presented by Dual.

The optimized structures of the most stable conformers of epoxides **5a** and **5b** are shown in Figure 8. Epoxide **5b** has a folded structure, while **5a** has a flatter structure. Despite the diastereomeric relationship, **5a** and **5b** have completely different conformations. That is, **5b** looks like a folded envelope, while **5a** is more like an open envelope. Considering that **5a** and **5b** are diastereomers, the difference in the biological activity of these compounds may be associated with their shape in space, which differentiates them in the biological target.

## 3. Materials and Methods

### 3.1. General

All compounds were purified by silica-gel column chromatography and recrystallization from diethyl ether. Reaction progress was followed by visualizing thin layer chromatography (TLC) plates with the spotted reaction mixture and starting material solution in an ultraviolet (UV) chamber [21]. The plates were then chemically stained with vanillin spray reagent.

Melting points were obtained on an MQAPF-301 (Microquímica Equipamentos Ltda, Palhoça, Brazil) melting point apparatus and were not corrected. Infrared (IR) spectra were acquired using a Varian 660-IR spectrophotometer (equipped with GLADI-ATR, Agilent, Santa Clara, CA, USA) with the attenuated total reflectance (ATR) method. NMR spectra were performed on a Brucker Avance DRX 400 MHz equipment (Billerica, MA, USA), using CDCl_3_ (Sigma-Aldrich, São Paulo, Brazil) as solvent. Chemical shifts were reported using tetramethylsilane (TMS) signals (δ = 0.0 ppm) as reference. The mass spectra were obtained on a Shimadzu equipment (Kyoto, Japan) GC-MS-QP5050A and GC-MS-QP2010 Ultra, after electron impact ionization (EI) at 70 eV. The NMR, IR, and mass spectra of the synthesized compounds can be found in the Appendix A.

### 3.2. Synthesis

#### 3.2.1. Synthesis of Amide (**2**)

Maleic anhydride (9.8 g, 0.1 mol) and isoprene (6.8 g, 0.1 mol) were irradiated in a microwave reactor for 5 min. After this period, the sealed tube was cooled to room temperature, providing a white solid that was washed with hexane to obtain anhydride (**1**) in 99% yield. Then, 3.0 mmol (500.0 mg) of the anhydride (**1**) was transferred to a round-bottomed flask and suspended in 3.0 mL of anhydrous dichloromethane (DCM). The mixture was stirred until complete solubilization. *m*-Anisidine (3.0 mmol) was added to the DCM solution, and the reaction mixture was stirred for 30 min. The solvent was removed under reduced pressure, and the residue was recrystallized using a mixture of hexane and DCM, providing amide (**2**) in 65% yield.

#### 3.2.2. Synthesis of 2-(3-methoxyphenyl)-5-methyl-3a,4,7,7a-tetrahydro-1H-isoindole-1,3(2H)-dione (**4**)

The amide (**2**, 0.500 mg) was dissolved in methanol (30 mL), and concentrated sulfuric acid (2.0 mL) was added dropwise to the solution. The mixture was stirred at room temperature for 1 h and transferred to a separation funnel. The solution was shaken with a mixture of DCM and saturated aqueous sodium bicarbonate (3 × 30 mL, 1:1 *v*/*v*). The organic phase was dried with anhydrous magnesium sulfate, filtered, and the solvent evaporated under reduced pressure. The residue was analyzed by GC/FID, and integration of the peak areas afforded a mixture containing 37% of the ester (**3**) and 63% of the imide (**4**). The crude reaction yield was 95%.

#### 3.2.3. Synthesis of Epoxides **5a** and **5b**

*meta*-Chloroperbenzoic acid (MCPBA, 0.43 g) was added to the crude mixture of compounds (**3** and **4**, 0.500 g) dissolved in DCM (10 mL). The mixture was stirred for 1 h and transferred to a separating funnel. The mixture was diluted with DCM (30 mL) and washed with saturated potassium bicarbonate solution (3 × 25 mL). The organic phase was dried with anhydrous magnesium sulfate, filtered, and concentrated under reduced pressure. The residue was purified by silica-gel column chromatography eluting with hexane/ethyl acetate (1:1) to afford **5a** and **5b,** which were further purified by recrystallization from diethyl ether. Compounds **5a** and **5b** were obtained in 19% and 66% yields, respectively. 

##### (1aS*,2aS*,5aR*,6aR*)-4-(3-methoxyphenyl)-1a-methylhexahydro-3H-oxireno[2,3-f]isoindole-3,5(4H)-dione (**5a**)

Yield 19%*,* white crystalline solid, TLC: *R*f 0.5 (hexane/ethyl acetate 1:1), m.p. 124.7–125.7 °C. ^1^H NMR (400 MHz, CDCl_3_) *δ*: 1.40 (3H, s, H1), 2.01 (1H, dd, *J* = 15 e 10 Hz, H2y), 2.08 (1H, dd, *J* = 15 e 10 Hz, H6y) 2.50 (1H, dd, *J* = 15 e 8 Hz, H2x), 2.68 (1H, ddd, *J* = 15, 8 e 4 Hz, H6x), 2.98–3.20 (3H, m, H2a, H5a e H6a), 3.80 (3H, s, H7′), 6.79 (1H, t, *J* = 2 Hz, H2′), 6.84 (1H, ddd, *J* = 8, 2 e 1 Hz, H6′), 6.93 (1H, ddd, *J* = 8, 2 e 1 Hz, H4′), 7.36 (1H, t, *J* = 8 Hz, H5′). ^13^C NMR (100 MHz, CDCl_3_) *δ*: 21.8 (C1), 23.8 (C6), 28.8 (C2), 35.3 (C2a), 36.9 (C5a), 55.0 (C1a), 55.4 (C7′), 56.1 (C6a), 112.2 (C2′), 114.5 (C4′), 118.6 (C6′), 129.9 (C5′), 132.7 (C1′), 160.1 (C3′), 178.5 (C3), 178.6 (C5). IR (solid) 3088, 2931, 1776, 1700, 1599, 1258, 790. MS (EI) *m/z* (%): 288 ([M + 1], 16), 287 ([M^+^.], 90), 244 (40), 149 (44), 123 (32), 81 (38), 53 (41), 43 (100).

##### (1aS*,2aR*,5aS*,6aR*)-4-(3-methoxyphenyl)-1a-methylhexahydro-3H-oxireno[2,3-f]isoindole-3,5(4H)-dione (**5b**)

Yield 66%*,* white crystalline solid, TLC: *R*f 0.2 (hexane/ethyl acetate 1:1), m.p. 126.2–127.4 °C. ^1^H NMR (400 MHz, CDCl_3_) *δ*: 1.32 (3H, s, H1), 2.18 (1H, dd, *J* = 15 e 7 Hz, H2y), 2.22 (1H, dd, *J* = 15 e 7 Hz, H6y), 2.59 (1H, d, *J* = 15 Hz, H2x), 2.77 (1H, dd, *J* = 15 e 4 Hz, H6x), 2.89 (2H, m, H2a e H5a), 3.04 (1H, d, *J* = 4 Hz, H6a), 3.80 (3H, s, H7′), 6.84 (1H, t, *J* = 2 Hz, H2′), 6.89 (1H, ddd, *J* = 8, 2 e 1 Hz, H6′), 6.92 (1H, ddd, *J* = 8, 2 e 1 Hz, H4′), 7.36 (1H, t, *J* = 8 Hz, H5′). ^13^C NMR (100 MHz, CDCl_3_) *δ*: 21.9 (C1), 23.6 (C6), 28.0 (C2), 35.3 (C2a), 36.7 (C5a), 55.4 (C7′), 56.7 (C1a), 57.6 (C6a), 112.4 (C2′), 114.6 (C4′), 119.0 (C6′), 129.8 (C5′), 133.8 (C1′), 160.1 (C3′), 179.6 (C3), 179.7 (C5). IR (solid) 3077, 2990, 2927, 1780, 1705, 1591, 1258, 849. MS (EI) *m/z* (%): 288 ([M + 1], 18), 287 ([M^+^.], 100), 244 (22), 204 (50), 149 (50), 93 (56), 43 (78).

### 3.3. Computation

The conformational searches were performed for candidate structures **5a** and **5b** (Figure 2) using Maestro 2018-1 (Maestro version 11.5.011) [22]. Eight conformers were found for structure (**5a**) and four conformers for structure (**5b**). The conformers were submitted to geometry optimization and frequency calculation using DFT at M06-2X/6-31+G(d,p) level of theory. Chemical shifts were obtained from the NMR shielding tensor values, which were computed for each candidate structure at B3LYP/6-311+G(2d,p) level in Gaussian 16 [18]. The shielding tensor of all nuclei was converted into referenced chemical shifts by subtracting the computed shielding tensors of TMS calculated at the same level of theory (^1^H = 31.8360; ^13^C = 183.8702). These operations were repeated for each candidate structure **5a** and **5b**. Solvation was addressed with the universal solvation model (smd) [23] during optimizations/frequencies and shielding tensors calculations.

Linear regression of the NMR chemical shifts of **5a_exp_** + **5b_exp_** vs. **5a_calc_** + **5b_calc_** and **5a_exp_** + **5b_exp_** vs. **5b_calc_** + **5a_calc_** was performed in MS Excel and transferred to the SI (Appendix A). With the values of the intercept and slope, the *δ*_scaled_ for each candidate structure was obtained. The experimental data set was compared with the calculated data after linear correction to determine the CMAE values, which are summarized in Table 3 (cf. green highlighted values) [20]. The CMAE analyses for each candidate structure were performed for the plot of the calculated (*δ*_calc_) chemical shifts against experimental (*δ*_exp_) chemical shifts, using the ^1^H and ^13^C NMR data.

Finally, the CMAE was calculated by the summation of the absolute value of ∆*δ* (∆*δ* = *δ*_exp_ − *δ*_calc_) of all N-nucleus and divided by the total number of N-nucleus. The calculated ^1^H and ^13^C chemical shifts using B3LYP/6-311+G(2d,p)//M06-2X/6-31+G(d,p) level of theory are presented in the Supporting Information. 

A goodness-of-fit probability was determined using the DP4 and CP3 methods described by Goodman [14,24]. DP4 analysis was accomplished by inputting computed and experimental chemical shifts into the DP4 analysis tool (located at http://www-jmg.ch.cam.ac.uk/tools/nmr/DP4/ (accessed on 28 January 2021)).

### 3.4. Biological Assay

The bioassays were performed with the synthesized substances **5a** and **5b**.

Lettuce seeds (*Lactuca sativa*) were placed on germination paper in a Petri dish (5.5 cm diameter). The solution containing the dissolved compound (2.5 mL) was added to the Petri dish, which was identified, sealed with plastic film, and transferred to the germination chamber (Bio-Oxygen Demand B.O.D.). The chamber was maintained at 25 °C in the absence of light for 120 h (5 days). The experiments were performed in triplicate with 15 seeds of each species. The experiments were conducted also with cucumber (*Cucumis sativus*), sorghum (*Sorghum bicolor*), and beggartick (*Bidens pilosa*) seeds.

Solutions of each tested substance were prepared in five distinct concentrations (500 μM, 300 μM, 150 μM, 100 μM, and 50 μM). The compounds were dissolved in dimethylsulfoxide (DMSO) and diluted with distilled water to prepare the most concentrated solution (500 μM) containing 0.3% DMSO. The remaining solutions were prepared from the 500 μM solution by the corresponding dilution with 0.3% aqueous DMSO [25,26,27].

The percentage of growth of shoots and roots was calculated according to the following formula:(1)$$G\left(\%\right)=\;\frac{\left(S-C\right)}{C}100$$
where *S* is the average length of the germinating shoots or roots of the plants, and *C* corresponds to the average growth of the negative control.

After the period of 5 days, the plates were frozen at 0 °C for 24 h to facilitate handling of the seedlings in the next step. They were then placed on a dark board and photographed. The aerial and root lengths of the seedlings were measured digitally.

The commercial herbicide Dual Gold (metolachlor) was used as a positive control, using 0.3% aqueous DMSO solution with the corresponding concentration (500 μM, 300 μM, 150 μM, 100 μM, and 50 μM). The seeds of the plants were obtained commercially, except beggartick seeds, which were collected at the campus of Universidade Federal de Viçosa, Viçosa, MG, Brazil.

## 4. Conclusions

The synthesis of a novel pair of epoxy derivatives from 5-methylhexahydrophtalimides (**5a** and **5b**) was carried out in five steps. The first step was a Diels–Alder reaction, followed by nucleophilic addition, esterification, cyclization, and finally epoxidation with MCPBA. The diastereomers obtained were isolated and characterized by the spectrometric methods, and **5b** was the major product. Epoxide **5b** was formed via attack of the MCPBA on the most hindered face of **4a** and **4a-ent,** which suggests that the steric effects are smaller than electronic effects in determining the outcome of this reaction.

Computational calculations followed by CMAE, CP3, and DP4 analyses were performed. The structures established by theoretical calculations were consistent with the structures found by exhaustive NMR interpretation.

Finally, biological tests were carried out with epoxides **5a** and **5b**, using lettuce, cucumber, sorghum, and beggartick seeds. The two tested substances interfered in the development of the seeds, and greater inhibition of growth was observed for the root part of the seeds. Substance **5a,** in general, presented better inhibition results in the development of both the root and aerial parts of the seeds than substance **5b**. Therefore, **5a** and **5b** proved to be suitable lead compounds for further investigation in the development of novel herbicides.

## Data Availability

The data presented in this study are available in this article and Appendix A.

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
