# Peer review of "Synthesis and Identification of Epoxy Derivatives of 5-Methylhexahydroisoindole-1,3-dione and Biological Evaluation"

_molecules, 2021, doi:10.3390/molecules26071923_

Round 1

Reviewer 1 Report

Torrent et al., reported a manuscript entitled “Synthesis and identification of epoxy derivatives of 5-methylhexahydroisoindole-1,3-dione and biological evaluation”. Authors were synthesized epoxy derivatives of 5-methylhexahydroisoindole1,3-diones, structures were determined by spectroscopic methods and shown herbicidal activity for these compounds. Although manuscript is written well, there is a more scope to improve the representation of the data.

  1. Manuscript needs to be check for typo errors and unique font style and size
  2. In Scheme 1, include reaction conditions with yields.
  3. Mass spec data is not clear, is it M+1?

Author Response

Reviewer 1:

Torrent et al., reported a manuscript entitled “Synthesis and identification of epoxy derivatives of 5-methylhexahydroisoindole-1,3-dione and biological evaluation”. Authors were synthesized epoxy derivatives of 5-methylhexahydroisoindole1,3-diones, structures were determined by spectroscopic methods and shown herbicidal activity for these compounds. Although manuscript is written well, there is a more scope to improve the representation of the data.

Response to Reviewer 1:

Thank you for reviewing our manuscript and for the suggestions to improve its overall quality. Please see below the responses for the questions raised.

Reviewer 1:

Manuscript needs to be check for typo errors and unique font style and size.

Response to Reviewer 1:

The manuscript has been checked. Font style, size and typo errors were corrected.

Reviewer 1:

In Scheme 1, include reaction conditions with yields.

Response to Reviewer 1:

Reaction conditions and yields were included in Scheme 1.

Reviewer 1:

Mass spec data is not clear, is it M+1?

Response to Reviewer 1:

The signal M+1 was also included in the mass spectra data because it was also present in the spectra. The values in the parentheses are the intensity of the signals. The original values reported in the manuscript are for M+ and not M+1. The odd number for M+ is because the compound contains a nitrogen atom.

Reviewer 2 Report

  1. S. Alvarenga et al reported “synthesis and identification of epoxy derivatives of 5-methylhexahydroisoindole-1,3-dione and biological evaluation”. In this work, the synthesis of a novel pair of epoxy derivatives from 5-methylhexahydrophtalimides was carried out in five steps. The diastereomers obtained were isolated and characterized by the spectrometric methods and 5b was the major product. Biological assays showed compounds 5a and 5b inhibited root growth of the weed Bidens pilosa by more than 70% at all the concentrations evaluated. Therefore, 5a and 5b proved to be substances with great herbicidal potential. I recommend publishing this manuscript in Molecules after a minor revision.
  2. In the introduction, Please illuminate the purpose of present work.
  3. On line 30, it appears that a comma is missing after the introductory phrase “in some situations”, consider adding a comma. The same error occurs in the line 201, and it is suggested to add a comma after the word “however”. Please review the full text and correct the same errors.
  4. On line 90, the noun phrase “assignment” seems to be missing a determiner before it. It is recommended to change to “the assignment”.
  5. On line 153, it appears that the preposition “without” is redundant. I suggest removing it.
  6. On line 187, the spelling of the word “beggartick” is not correct, please correct it to “beggarticks”. On line 357, “stablished” should be amended to “established”.
  7. On line 190, the plural verb “were” does not appear to agree with the singular subject development. Consider changing the verb form for subject-verb agreement. It should be amended to “was”. Please review the full text and correct the same errors.
  8. Please pay attention to the writing of the units “µM” in the text, “µ” should be italic, check the full text to correct it.
  9. Please check the full text carefully and pay attention to the writing format. For example, on line 94、126, “Table 1、3” should not be italic, please keep the same format as others. Also there is an error in the order of the secondary headings. I suggest that it should be corrected.

Author Response

Reviewer 2:

  1. Alvarenga et al reported “synthesis and identification of epoxy derivatives of 5-methylhexahydroisoindole-1,3-dione and biological evaluation”. In this work, the synthesis of a novel pair of epoxy derivatives from 5-methylhexahydrophtalimides was carried out in five steps. The diastereomers obtained were isolated and characterized by the spectrometric methods and 5b was the major product. Biological assays showed compounds 5a and 5b inhibited root growth of the weed Bidens pilosa by more than 70% at all the concentrations evaluated. Therefore, 5a and 5b proved to be substances with great herbicidal potential. I recommend publishing this manuscript in Molecules after a minor revision.

Response to Reviewer 2:

Thank you for reviewing our manuscript and for the suggestions to improve its overall quality. Please see below the responses for the questions raised. The suggested corrections were highlighted in yellow in the main text.

Reviewer 2:

In the introduction, please illuminate the purpose of present work.

Response to Reviewer 2:

The following paragraphs were included (or modified) in the introduction to illuminate the purpose of the work.

“In 2050, it is estimated that the global population will reach the mark of 9.1 billion people. As a result, world food production will need to increase by 70–100%. Weeds are agricultural pests and cause the highest percentage of loss of income from food production. They compete with the plantations for sunlight, water, and nutrients, harbor insects and pathogens (fungi, bacteria, and viruses) promoting losses in production. In addition, weeds destroy native habitats, threatening plants and animals in the local ecosystem.”

“Therefore, we decided to use the known Diels-Alder reaction to prepare the tetrahydrophthalimide (4) in three steps, which was converted into the epoxy compounds (5a and 5b). From this reaction, we obtained two novel pairs of diastereomers, whose relative configurations were determined by comparison of the experimental with the computed NMR chemical shifts.”

“In this context, we have prepared two novel epoxy compounds derived from 5-methylhexahydroisoindole-1,3-dione from the Diels-Alder reaction between maleic anhydride and isoprene. The diastereoisomers had their structures elucidated by NMR, and their relative configurations were confirmed by comparing the experimental with the calculated NMR chemical shifts. The herbicidal activities of these compounds were evaluated against seeds of Bidens pilosa L., Cucumis sativus L., Lactuca sativa L., and Sorghum bicolor L.”

Reviewer 2:

On line 30, it appears that a comma is missing after the introductory phrase “in some situations”, consider adding a comma. The same error occurs in the line 201, and it is suggested to add a comma after the word “however”. Please review the full text and correct the same errors.

Response to Reviewer 2:

Commas were included as suggested.

Reviewer 2:

On line 90, the noun phrase “assignment” seems to be missing a determiner before it. It is recommended to change to “the assignment”.

Response to Reviewer 2:

It was corrected as suggested.

Reviewer 2:

On line 153, it appears that the preposition “without” is redundant. I suggest removing it.

Response to Reviewer 2:

It was corrected as suggested.

Reviewer 2:

On line 187, the spelling of the word “beggartick” is not correct, please correct it to “beggarticks”.

Response to Reviewer 2:

Beggartick is the singular name and beggarticks is the plural name. For this occasion, beggartick is appropriate, because the other names of the seeds were also in the singular.

Reviewer 2:

On line 357, “stablished” should be amended to “established”.

Response to Reviewer 2:

It was corrected as suggested.

Reviewer 2:

On line 190, the plural verb “were” does not appear to agree with the singular subject development. Consider changing the verb form for subject-verb agreement. It should be amended to “was”. Please review the full text and correct the same errors.

Response to Reviewer 2:

This sentence was rephrased:

“The values of inhibition and stimulation of the substances were evaluated according to the growth (positive values) or inhibition (negative values) of the root and stem of the tested seeds.”

Reviewer 2:

Please pay attention to the writing of the units “µM” in the text, “µ” should be italic, check the full text to correct it.

Response to Reviewer 2:

“µ” was italicized as suggested.

Reviewer 2:

Please check the full text carefully and pay attention to the writing format. For example, on line 94、126, “Table 1、3” should not be italic, please keep the same format as others. Also there is an error in the order of the secondary headings. I suggest that it should be corrected.

Response to Reviewer 2:

It was corrected as suggested.

Reviewer 3 Report

In this manuscript, the authors have reported the synthesis of epoxide derivatives of Hexahydropthalimide derivatives. Further, they confirmed the spacial orientation of the 2 epoxides using 1D NMR, 2D NMR, and computational studies. Finally, both the epoxides prepared were tested for their herbicidal potential. 

Here are my comments:

1) Please provide at least one specific examples, and their structures of important cyclic imides (as claimed in page 1, lines 24-25).

2) Please introduce CP3, and DP4 before using the abbreviations. A couple of sentences defining them should suffice. 

3) Please cleanup the Chemdraw structures. Make sure the bond angles are correct. Also, mention the reaction conditions on the arrows in Scheme 1.

4) Please assign different numberings for Figures and Tables in SI. Figure S1 and Table S1 is confusing to the readers.

5)  Please represent the Biological assay bar diagrams using different distinct colors for different concentrations of the testing compounds. The colors chosen (different shades of black) is hard to understand.

6) It will be really helpful for the readers, if authors introduce the positive control herbicide, and add the structure of Dual gold (Metolachlor). 

7) Based on the biological data presented, it is clear that these epoxide derivatives have some herbicidal activity. There is not enough evidence to claim these as great potential herbicides. Moreover, authors do not offer an explanation for non-dose responsive activity (including the control), and sometimes the compounds resulted in the growth of the seeds rather than inhibition. Please shed some light on the abnormalities in growth and inhibition.

To conclude, the chemistry involved in the synthesis is well known and involves well established reactions. The compounds 5-Methyhexahydroisoindole-1,3-dione derivatives are known in the literature, and are commercially available. Given the low originality / novelty, and less interest in the scientific community I recommend authors to publish this work in other suitable journals. Else, it can be reconsidered after inclusion of an SAR study of more epoxide derivatives with different substituents on the phenyl ring of pthalimides. 

Author Response

Reviewer 3:

In this manuscript, the authors have reported the synthesis of epoxide derivatives of Hexahydropthalimide derivatives. Further, they confirmed the spacial orientation of the 2 epoxides using 1D NMR, 2D NMR, and computational studies. Finally, both the epoxides prepared were tested for their herbicidal potential.

Response to Reviewer 3:

Thank you for reviewing our manuscript and for the suggestions to improve its overall quality. Please see below the responses for the questions raised. The suggested corrections were highlighted in yellow in the main text.

Reviewer 3:

Here are my comments:

1) Please provide at least one specific examples, and their structures of important cyclic imides (as claimed in page 1, lines 24-25).

Response to Reviewer 3:

Figure 1, with examples of commercial herbicides containing cyclic imides, was included in the introduction.

Reviewer 3:

2) Please introduce CP3, and DP4 before using the abbreviations. A couple of sentences defining them should suffice.

Response to Reviewer 3:

The following paragraph in the introduction describes the statistical methods MAE, CP3 and DP4:

“Statistical methods commonly used in structure determination of organic compounds are mean absolute error (MAE), CP3 and DP4 probabilities. DP4 tool is applied when there are experimental data for one compound which is compared to two or more candidate structures, whereas CP3 is used when NMR chemical shifts are available for two or more substances. The calculation of DP4 can be done at http://www.jmg.ch.cam.ac.uk/tools/nmr/DP4 for assigning one set of experimental data to one of many candidate structures. The calculation of CP3 can be done at http://www-jmg.ch.cam.ac.uk/tools/nmr/CP3.html for assigning two or more set of experimental data to one of many potential structures. The probability that the assignment combination is correct (5aexp = 5acalc, 5bexp = 5bcalc) is calculated by the Bayes' theorem (Equation S1, SI). [13–16]”

Reviewer 3:

3) Please cleanup the Chemdraw structures. Make sure the bond angles are correct. Also, mention the reaction conditions on the arrows in Scheme 1.

Response to Reviewer 3:

The structures were cleaned up in Chemdraw as suggested.

Reviewer 3:

4) Please assign different numberings for Figures and Tables in SI. Figure S1 and Table S1 is confusing to the readers.

Response to Reviewer 3:

The structures in the main manuscript which were reproduced in the SI were given the same numbers. The “Figures and Tables” which are different from the “Figures and Tables” in the main manuscript were given different numbers. For example, Figure 1 in the SI was labeled Figure S1. This was done to differentiate from the Figure in the main manuscript. The S in the name stands for Supplementary. Therefore, whenever “S” is detected in the name, the “Table and Figure” will be found in the Supplementary Information.

Reviewer 3:

5)  Please represent the Biological assay bar diagrams using different distinct colors for different concentrations of the testing compounds. The colors chosen (different shades of black) is hard to understand.

Response to Reviewer 3:

The colors of the bar diagrams in the Figures of the Biological assays were modified as suggested.

Reviewer 3:

6) It will be really helpful for the readers, if authors introduce the positive control herbicide, and add the structure of Dual gold (Metolachlor).

Response to Reviewer 3:

The structure of metolachlor and the following paragraph were included in the main text,

“The values of inhibition and stimulation of the substances were evaluated according to the growth (positive values) or inhibition (negative values) of the root and stem of the tested seeds. Dual (Syngenta® Company, São Paulo, Brazil; S-metolachlor, Figure 3) was used as positive control and aqueous solution of 0.3% DMSO (v/v) as negative control.”

Reviewer 3:

7) Based on the biological data presented, it is clear that these epoxide derivatives have some herbicidal activity. There is not enough evidence to claim these as great potential herbicides. Moreover, authors do not offer an explanation for non-dose responsive activity (including the control), and sometimes the compounds resulted in the growth of the seeds rather than inhibition. Please shed some light on the abnormalities in growth and inhibition.

Response to Reviewer 3:

The positive control used was Dual and the negative control was aqueous solution of 0.3% DMSO (v/v). The following paragraph was included in the main text:

“The values of inhibition and stimulation of the substances were evaluated according to the growth (positive values) or inhibition (negative values) of the root and stem of the tested seeds. Dual (Syngenta® Company, São Paulo, Brazil; S-metolachlor, Figure 3) was used as positive control and aqueous solution of 0.3% DMSO (v/v) as negative control.”

The following paragraph was included in the “Biological Assay” section:

“The percentage of growth of shoots and roots was calculated according to the following equation:

Esquation S1

where S is the average length of the germinating shoots or roots of the plants, and C corresponds to the average growth of the negative control.”

The last sentence in the conclusion was rephrased:

“Therefore, 5a and 5b proved to be substances with great herbicidal potential.” (original sentence)

“Therefore, 5a and 5b proved to be suitable lead compounds for further investigation in the development of novel herbicides.” (modified sentence)

Imide 5b was less active than 5a and for some seeds 5b presented stimulus of growth of the seeds. However, 5b should not be considered plant hormone, because the growth stimulus was only marginal. Therefore, we have not discussed the hormone potential for substance 5b.

Reviewer 3:

To conclude, the chemistry involved in the synthesis is well known and involves well established reactions. The compounds 5-Methyhexahydroisoindole-1,3-dione derivatives are known in the literature, and are commercially available. Given the low originality / novelty, and less interest in the scientific community I recommend authors to publish this work in other suitable journals. Else, it can be reconsidered after inclusion of an SAR study of more epoxide derivatives with different substituents on the phenyl ring of pthalimides.

Response to Reviewer 3:

The compound 5-methyhexahydroisoindole-1,3-dione and its derivatives are known. Figure 1, which was included in revised version of the manuscript, presents some of these compounds. In the present work we present the synthesis of novel imides and determine the relative stereochemistry of the four chiral centers by the spectrometric methods assisted by theoretical calculations. Besides that, biological assays were carried out. The syntheses of novel imides and epoxides are in progress and will be published in due course.

Round 2

Reviewer 3 Report

Thanks for the corrections.

I am happy to recommend the editors to accept this manuscriprt for publication in Molecules.